# The Statin Target Hmgcr Regulates Energy Metabolism and Food Intake through Central Mechanisms

**DOI:** 10.3390/cells11060970

**Published:** 2022-03-11

**Authors:** Michael J. Williams, Ahmed M. Alsehli, Sarah N. Gartner, Laura E. Clemensson, Sifang Liao, Anders Eriksson, Kiriana Isgrove, Lina Thelander, Zaid Khan, Pavel M. Itskov, Thiago C. Moulin, Valerie Ambrosi, Mohamed H. Al-Sabri, Francisco Alejandro Lagunas-Rangel, Pawel K. Olszewski, Helgi B. Schiöth

**Affiliations:** 1Functional Pharmacology and Neuroscience, Department of Surgical Sciences, Uppsala University, 751 24 Uppsala, Sweden; michael.williams@neuro.uu.se (M.J.W.); ahmedm.alsheli@neuro.uu.se (A.M.A.); laura.clemensson@neuro.uu.se (L.E.C.); sifang.liao@neuro.uu.se (S.L.); eriksson.erik.anders@gmail.com (A.E.); thelanderlina@gmail.com (L.T.); zaid.khan@slu.se (Z.K.); itskovpa@gmail.com (P.M.I.); thiago.moulin@neuro.uu.se (T.C.M.); val.ambrosi@googlemail.com (V.A.); mohamed.alsabri@neuro.uu.se (M.H.A.-S.); francisco.lagunas@neuro.uu.se (F.A.L.-R.); 2Faculty of Medicine, King Abdulaziz University and Hospital, Al Ehtifalat St., Jeddah 21589, Saudi Arabia; 3Department of Biological Sciences, University of Waikato, Private Bag 3105, Hamilton 3240, New Zealand; sarahgartner1@gmail.com (S.N.G.); k_isgrove@outlook.co.nz (K.I.); pawel.olszewski@waikato.ac.nz (P.K.O.); 4Department of Plant Protection Biology, Swedish University of Agricultural Sciences (SLU), Sundsvägen 14, 230 53 Alnarp, Sweden

**Keywords:** body maintenance index, obesity, statins, mevalonate pathway, metabolism, feeding behavior, hypothalamus

## Abstract

The statin drug target, 3-hydroxy-3-methylglutaryl-CoA reductase (HMGCR), is strongly linked to body mass index (BMI), yet how HMGCR influences BMI is not understood. In mammals, studies of peripheral HMGCR have not clearly identified a role in BMI maintenance and, despite considerable central nervous system expression, a function for central HMGCR has not been determined. Similar to mammals, Hmgcr is highly expressed in the *Drosophila melanogaster* brain. Therefore, genetic and pharmacological studies were performed to identify how central *Hmgcr* regulates *Drosophila* energy metabolism and feeding behavior. We found that inhibiting *Hmgcr*, in insulin-producing cells of the *Drosophila pars intercerebralis* (PI), the fly hypothalamic equivalent, significantly reduces the expression of insulin-like peptides, severely decreasing insulin signaling. In fact, reducing *Hmgcr* expression throughout development causes decreased body size, increased lipid storage, hyperglycemia, and hyperphagia. Furthermore, the *Hmgcr* induced hyperphagia phenotype requires a conserved insulin-regulated α-glucosidase, *target of brain insulin* (*tobi*). In rats and mice, acute inhibition of hypothalamic Hmgcr activity stimulates food intake. This study presents evidence of how central Hmgcr regulation of metabolism and food intake could influence BMI.

## 1. Introduction

Obesity has become an international problem, leading the World Health Organization (WHO) to declare it a global pandemic [1]. With its associated health problems, including type 2 diabetes, cardiovascular diseases, and some forms of cancer, obesity is correlated with decreased life expectancy. Even though obesity is associated with serious diseases, as a non-communicable disease (NCD), it has not received the attention of other rapidly spreading infectious diseases. Although lifestyle influences, such as increased food availability and intake, along with decreased physical activity, have significantly increased the frequency of obesity, genome-wide association (GWA) studies suggest a substantial genetic contribution. Recent estimates indicate that heritable contributions may account for as much as 60% of the body mass index (BMI) variation observed among the population. In fact, GWA studies have linked more than 90 genes to increased BMI and obesity [2,3,4].

HMGCR, which catalyzes an early rate-limiting step of cholesterol biosynthesis [5], has been linked to the regulation of BMI in multiple GWA studies [2,3]. Statins, inhibitors of HMGCR, are well documented as being effective in the treatment against hypercholesterolemia and in decreasing the risk of cardiovascular disease morbidity and mortality, by reducing plasma levels of low-density lipoprotein-cholesterol (LDL-C), while increasing beneficial high-density lipoprotein-cholesterol (HDL-C) concentrations [6,7]. However, in addition to the positive effects of statins, population-based studies have reported that long-term use is associated with an increased risk for type 2 diabetes [8,9]. In fact, a large meta-analysis of two HMGCR single nucleotide polymorphisms (SNPs), including 43 studies with 223,463 individuals, and randomized controlled trials of statin treatment in 129,170 participants, concluded that the HMGCR SNPs and statin treatment were similarly associated with lowering LDL-C concentration while raising the risk for increased BMI and type 2 diabetes [10].

Although multiple GWA studies have linked HMGCR to the regulation of BMI, reports on the peripheral activity of HMGCR have not clearly linked the enzyme to BMI maintenance [11,12]. Furthermore, even though HMGCR is highly expressed in the brain [13,14], no studies have addressed the specific role of this gene in relation to the central regulation of energy metabolism or BMI maintenance. There is also evidence from studies using statins that central HMGCR activity may play a role in BMI maintenance. All statin drugs tested, both hydrophilic and lipophilic, have been detected in the mouse cerebral cortex, where they were shown to modify the expression of multiple genes, including those known to regulate metabolism, *insulin-like growth factor-binding protein 3* (IGFBP3) and *glucose-6-phosphate isomerase* (GPI), and feeding behavior, *neuropeptide Y receptor Y1* (NPY1R) and *calcium voltage-gated channel subunit alpha1 G* (CACNA1G) [15,16]. 

Impairment of Hmgcr activity caused by genetic variants or statin exposure has been shown to affect insulin secretion and promote the development of diabetes [9,10]. Insulin is an important regulator of metabolism, growth, reproduction and life expectancy. In adult flies, only a set of medium-sized neurosecretory cells in the *pars intercerebralis* (PI) of the brain, called insulin-producing cells (IPCs), produce and secrete the insulin-like peptides (ILPs) DILP2, 3, and 5 [17]. These cells share functional and physiological similarities with the β-cells of the mammalian pancreas. Through transcriptome analysis of IPCs, an enrichment of mRNAs was found that are orthologs of key enzymes for insulin processing in mammals, as well as for components of dense-core vesicles and signal transducers that control insulin secretion, such as ammon, ia2, and Pkc98E mRNAs [18]. DILPs are released from axon endings in the hemolymph, corpora cardiac, *corpora allata*, crop, and foregut to distribute throughout the insect body and carry out their physiological functions [17,18]. Similar to mammals, *Hmgcr* is highly expressed in the brain of *Drosophila melanogaster* and has been shown to signal downstream of insulin signaling to regulate juvenile hormone-induced sexual dimorphism [14]. In the present study, we use the *Drosophila* model to identify a function for Hmgcr in the central IPCs and show how it relates to energy metabolism and feeding behavior. Furthermore, to corroborate whether what was found in *Drosophila* also occurs in mammals, we examined the regulation of feeding behavior by simvastatin in the hypothalamus of mice and rats.

## 2. Materials and Methods

### 2.1. Animal Ethics

Experimental procedures were carried out in accordance with the guidelines of the Federation of European Laboratory Animal Science Associations, based on European Union legislation (Directive 2010/63/EU), as well as NIH Guide for the Care and Use of Laboratory Animals (NIH Publ., no. 80–23, rev. 1996), local laws and policies. The mouse procedures were approved by the local ethics committee in Uppsala, Sweden, while the rat procedures were approved by the institutional ethics committee at the University of Waikato. Procedures on *Drosophila* do not require bioethical authorization.

Ethics Committee Name: Uppsala Djurförsöksetiska Nämnd

Approval Code: C 71/14

Approval Date: 25 April 2014

Ethics Committee Name: University of Waikato Animal Ethics Committee

Approval Code: Protocol #989

Approval date: 1 April 2016

### 2.2. Drosophila Studies

#### 2.2.1. Fly Husbandry

Fly stocks were maintained at 25 °C with 60% relative humidity in a 12:12 h light:dark cycle. Unless otherwise stated, *Drosophila* stocks were maintained with Jazz-Mix Drosophila food (Thermo-Fisher Scientific, Göteborg, Sweden) supplemented with yeast extract (58 g/dL sugar:12 g/dL protein, referred to as high-sugar food). The following strains were used to knockdown Hmgcr: *w^1118^*, *UAS-Hmgcr^RNAi1^* (*P{KK101807}VIE-260B*) from the Vienna *Drosophila* RNAi Centre (VDRC, Vienna, Austria) and *w^1118^*, *UAS-Hmgcr^RNAi2^* (*P{UAS-RNAi-HMGCR}10367-R3*) from the National Institute of Genetics stock center (NIG, Mishima, Japan). Likewise, flies were used with the following GAL4 drivers: *y[1] v[1];P{GawB}elavC155 w** (referred to as *elav-GAL4*, containing a neuron-specific promoter); *w*; P{GawB}Aug21/CyO* (referred to as *Aug21-GAL4*, containing a *corpus allatum*-specific promoter), *P{w[+mC] = Ilp2-GAL4.R}EQU2* (referred to as *Dilp2-GAL4*, containing *a promoter specific for insulin-producing cells (IPCs)*), all from the Bloomington Stock Center (Bloomington, IN, USA). The following strain was used to knockdown Fpps: *ry[506], P{ry[+t7.2] = PZ}Hmgcr[01152]/TM3, ry[RK] Sb[1] Ser[1]; y[1] sc[*] v[1] sev[21]; P{y[+t7.7] v[+t1.8] = TRiP.HMC05747}attP40 (UAS-FppsRNAi);* and the following strain was used to knockdown Tobi: *P{w[+mW.hs] = GawB}48Y* and *y1 v1; P{y[+t7.7] v[+t1.8] = TRiP.HMJ02101}attP40* (*UAS-tobi^RNAi^*), both from the Bloomington Stock Center (Bloomington, IN, USA). All transgenic strains were initially crossed into the same *w^1118^* genetic background. Genetic crosses: Virgin female *elav-GAL4, Dilp2-GAL4, Aug21-GAL4* or *48Y-GAL4* were crossed with male *UAS-RNAi* or overexpression lines. Two different controls were used: virgin female *w^1118^* crossed with male UAS lines, and virgin female GAL4 lines crossed with male *w^1118^*. The *UAS-tobi* overexpression line (referred to as *tobi^OE^*) was a gift from Dr. Michael Pankratz [19]. Adult flies were collected immediately after they eclosed and placed at 29 °C for 5–7 days, to ensure maximum expression of the UAS line, before any experiments were performed. Unless otherwise stated, experiments were performed at 25 °C.

#### 2.2.2. Macronutrient Diets

All diets consisted of varying concentrations (g/dL) of sucrose (Sigma, Malmö, Sweden) or yeast extract (VWR, Karlskoga, Sweden) in 1% agarose. Low-sugar food is 10 g/dL sucrose and 10 g/dL yeast. Other diets used were 2.5 g/dL sugar:2.5 g/dL yeast, 40 g/dL sugar:40 g/dL yeast, 40 g/dL sugar:10 g/dL yeast, and 10 g/dL sugar:40 g/dL yeast. Newly eclosed adult males were maintained on these diets for five days at 29 °C, 50% humidity on a 12:12 h light:dark cycle.

#### 2.2.3. RNA Extraction

To obtain equivalent amounts of RNA, RNA was extracted by homogenizing either 25 male fly heads (for quantitative RT-PCR of *Hmgcr, Ilp2, Ilp3,* and *Ilp5*) or 10 male fly bodies (for quantitative RT-PCR of *Adipokinetic Hormone, Akh*) in PBS, all equally aged to between 5–7 days post-eclosion. An equal volume of phenol:choloroform:isoamyl alcohol solution (25:24:1) (Sigma-Aldrich, Malmö, Sweden) was added to the homogenized flies and mixed. The solution was centrifuged for 5 min at 12,000× *g* at room temperature. The aqueous phase was transferred to a new tube and an equal volume of chloroform was added, followed by centrifugation at the previously mentioned speed and time. An ethanol and silica suspension (1 g/mL) (Sigma, Malmö, Sweden) was added to the aqueous phase and incubated for 1 min before additional centrifugation. The pellet was washed with 70% ethanol and let dry. To remove any DNA, the sample was treated with DNAse (Thermo-Fisher Scientific, Göteborg, Sweden) for 30 min at 37 °C and subsequently, at 65 °C for 15 min. The pellet was resolved in 50 µL DEPC-H2O and incubated for 5 min. To remove the silica suspension, the sample was centrifuged, and the RNA solution was transferred to a new tube. A spectrophotometer (model ND-1000, Nanodrop) was used to measure total RNA concentration.

#### 2.2.4. cDNA Synthesis

The High-capacity RNA-to-cDNA Kit (Applied Biosystems, Stockholm, Sweden) was used for cDNA synthesis and performed according to the manufacturer’s instructions.

#### 2.2.5. Quantitative RT-PCR (qPCR)

Relative expression levels of three housekeeping genes (*EF-1, RpL32*, & *RpL11*) and of the genes of interest were determined with quantitative RT-PCR (qPCR). Each reaction, with a total volume of 20 µL, contained 20 mM Tris/HCl pH 9.0 (Thermo-Fisher Scientific, Göteborg, Sweden), 50 mM KCl (Thermo-Fisher Scientific, Göteborg, Sweden), 4 mM MgCl_2_ (Thermo-Fisher Scientific, Göteborg, Sweden), 0.2 mM dNTP (Thermo-Fisher Scientific, Göteborg, Sweden), DMSO (1:20) (Thermo-Fisher Scientific, Göteborg, Sweden) and SYBR Green (1:50,000) (Thermo-Fisher Scientific, Göteborg, Sweden). Template concentration was 5 ng/µL and the concentration of each primer was 2 pmol/µL. Primers were designed with Beacon Designer (Premier Biosoft, San Francisco, CA, USA) using the SYBR Green settings. All qPCR experiments were performed in duplicates; for each primer pair, a negative control with water and a positive control with 5 ng/µL of genomic DNA were included on each plate. Amplifications were performed with 0.02 µg/mL Taq DNA polymerase (Biotools, Stockholm, Sweden) under the following conditions: initial denaturation at 95 °C for 3 min, 50 cycles of denaturing at 95 °C for 15 s, annealing at 52.8–60.1 °C for 15 s, and extension at 72 °C for 30 s. Analysis of qPCR data was performed using MyIQ 1.0 software (Bio-Rad). Primer efficiencies were calculated using LinRegPCR3 and samples were corrected for differences in primer efficiencies. The GeNorm protocol described by Vandesompele et al. [20] was used to calculate normalization factors from the expression levels of the housekeeping genes. Differences in gene expression between groups were analyzed with ANOVA followed by Fisher’s PLSD test where appropriate. *p* < 0.05 was used as the criterion of statistical significance. The following primers were used: *EF-1* F: 5′-GCGTGGGTTTGTGATCAGTT-3′, R: 5′-GATCTTCTCCTTGCCCATCC-3′; *RpL32* F: 5′-CACACCAAATCTTACAAAATGTGTGA-3′, R: 5′-AATCCGGCCTTGCACATG-3′; *RpL11* F: 5′-CCATCGGTATCTATGGTCTGGA-3′, R: 5′-CATCGTATTTCTGCTGGAACCA-3′; *Hmgcr* F: 5′-CCTGAATGTGAGCAATAATC-3′, R: 5′-TAACTACCAAGGCGATGA-3′; *Ilp2* F: 5′-AGCCTTTGTCCTTCATCT-3′, R: 5′-CATACTCAGCACCTCGTT-3′; *Ilp3* F: 5′-AAACTGCCCGAAACTCTC-3′, R: 5′-AGCATCTGAACCGAACTAT-3′; *Ilp5* F: 5′-CTTGATGGACATGCAGAG-3′, R: 5′-GAAAAGGAACACGATTTG-3′; *Akh* F: 5′-CTGGTCCTGGAACCTTTT-3′, R: 5′-GAGCTGTGCCTGAGATTG-3′.

#### 2.2.6. Library Preparation and Sequencing

RNA-seq reads for the entire transcriptome were obtained using SOLiD 5500xl paired-end sequencing from Life Technologies. Initial quality analysis was performed using a proprietary ‘XSQ Toolkit’ provided by Life Technologies. Further analysis was performed using the ‘Tuxedo suit’ [21,22], mainly composed of three tools: TopHat, Cufflinks, and CummRbund. Reads were then aligned to the *D. melanogaster* reference genome (build dmel_r5.47_FB2012_05) obtained from flybase using TopHat with the pre-built bowtie index downloaded from the TopHat home page (http://ccb.jhu.edu/software/tophat/index.shtml, accessed on 10 January 2022). Transcript assembly and abundance estimation was estimated using Cufflinks v2.0.2. Subsequently, differential expression tests were performed using cuffcompare and cuffdiff. The calculated *p* and q values (the FDR-adjusted *p* value of the test statistic) from cuffdiff were used to determine the significance of differential expression.

#### 2.2.7. Immunohistochemistry

Protocol for the immunohistochemical staining of the adult brain of *Drosophila melanogaster*: equally aged male *ry[506], P{ry[+t7.2] = PZ}Hmgcr[01152]/TM3, ry[RK] Sb[1] Ser[1]* flies, 5–7 days old, were decapitated under anesthesia and fixated in a staining glass bowl containing 4% Formaldehyde/1X PBS solution (methanol-free) (Thermo-Fisher Scientific, Sweden) with 12.5% Tween-20 (Sigma, Sweden) on ice and agitation for 1 h. After fixation, heads were washed four times for 15 min with 0.1 M sodium phosphate buffer (PBS) (Sigma, Sweden) at room temperature. Brains were dissected on a silicon plate and again washed four times for 15 min with 0.1 M PBS on room temperature. Tissues were blocked with 10% normal goat serum (NGS) in 0.01 M PBS with 0.25% Triton X (PBS-tx) for 45 min at 25 °C. NGS was discarded and tissues incubated with primary antibodies (α-β-Gal (Sigma, Stockholm, Sweden) derived from rat and α-Dilp2 (gift from Roel Nusse—derived from chicken) diluted 1:2800 and 1:2500 in 10% NGS in 0.01 M PBS-tx overnight at 4°C. Afterwards, brains were washed again five times with 0.01 M PBS-tx for 15 min at 25 °C and incubated with secondary antibody (α-rat Alexa 488 and α-chicken Alexa 594 (Abcam, Cambridge, UK)) diluted 1:800 in 0.01 M PBS-tx overnight at 4 °C. Again, tissues were washed three times for 15 min with 0.01 M PBS-tx and mounted with 60% glycerol (Sigma, Sweden) containing 1.6% propyl gallate (Sigma, Sweden). Images were acquired with a 63X oil immersion objective on an inverse confocal microscope (Zeiss LSM 510 META) running on Zeiss LSM 510 software (version 4.2 SP1).

#### 2.2.8. Western Blot Assay

Five heads of male flies with 5 to 7 days of age were homogenized in 2X Laemmli sample buffer (Bio-Rad, Hercules, CA, USA) and equal amounts of homogenate were loaded onto SDS-PAGE gels and blotted according to standard protocol. Blots were probed with the following antibodies (all at 1:2000 dilution): (1) phospho-*Drosophila* Akt (Ser505) Antibody (Cell Signaling #4054), (2) Akt (pan) (C67E7) Rabbit mAb (Cell Signaling #4691), (3) E7 anti-beta-tubulin (Developmental Studies Hybridoma Bank). At least four experiments were performed, and multiple exposures were taken; Western blot quantification was done on lightly exposed blots using Licor Image analysis software (LI-COR Biosciences—GmbH). Blots were stripped and re-probed using the mild stripping protocol from Abcam (http://www.abcam.com/ps/pdf/protocols/stripping%20for%20reprobing.pdf, accessed on 10 August 2021). Primary antibodies were used in the following order—P-Akt, total Akt, β-tubulin. Secondary antibodies conjugated to HRP (Genescript) were used, and the signals were detected by chemiluminescence using the Enhanced ECL kit (Biorad).

#### 2.2.9. Oxidative Stress Assay

Resistance to oxidative stress was performed according to [23]. In short, 5–7-day-old male flies previously maintained on either a high-sugar diet (58 g/dL sugar:12 g/dL yeast) or a low-sugar diet (10 g/dL sugar:10 g/dL yeast) were transferred into glass vials containing 1% agar, 5% sugar and 20 mM paraquat (Sigma, St. Louis, MO, USA) [23]. Dead flies were counted twice a day. For both assays, a total of 50 flies were used for each genotype and parameter (10 flies per vial). Survival differences were analyzed by proportional hazard analysis.

#### 2.2.10. CAFE Assay

A vial, 9 cm by 2 cm (height by diameter), containing 1% agarose (5 cm high) (Invitrogen, Göteborg, Sweden) to provide moisture and humidity for the flies, was used for this assay [24]. A calibrated capillary glass tube (5 µL, VWR International) was filled with liquid food which contains 5% sucrose (Sigma, Sweden), 5% yeast extract (Sigma, Sweden), and 0.5% food-coloring dye. A layer of mineral oil was used to prevent the liquid food from evaporating. Five males, which were 5–7 days old, were put inside the chamber and the opening of the vial was covered with paraffin tape, with a capillary tube being inserted from the top through the tape. The experimental setup was kept at 25 °C, 50% humidity on a 12:12 h light:dark cycle. At least 10 replicates were performed for each genotype. All male flies were equally aged to 5–7 days post-eclosion.

#### 2.2.11. Triglyceride Assay

The 5–7 day old flies (25 males) were homogenized with 100 μL of PBST buffer (1X PBS with 10% Tween 20), incubated at 70 °C for 5 min, and then centrifuged at maximum speed for 10 min. The supernatant was transferred into a new microcentrifuge tube and used as samples. The glycerol standard (Sigma, Stockholm, Sweden) was used to generate a standard curve with concentrations of 1.0, 0.8, 0.6, 0.4, and 0.2 mg/mL equivalent triolein concentration. One hundred μL of free glycerol reagent was added with 10 μL of blank (PBST), standards or samples, and initial absorbance at 540 nm was measured after incubation at 37 °C for 15 min. The concentration of free glycerol in the samples was calculated from a standard curve generated by these initial absorbance values. Then, 20 μL of triglyceride reagent was added to each standard and samples and incubated at 37 °C for 15 min. The final absorbance measurement was taken at 540 nm to calculate triglyceride concentrations from the generated standard curve. The protein concentration of each sample was measured with the Bio-Rad Protein Assay Kit (Bio-Rad, Hercules, CA, USA). The concentrations of free glycerides and triglycerides in samples (mg per mg of protein) were calculated from 10 replicates.

#### 2.2.12. Trehalose and Glucose Assays

Four substrates were measured: circulating trehalose, stored trehalose, circulating glucose, and glycogen. All extracted substrates were converted to a glucose solution for final analysis via spectrophotometry. Male flies, aged 5–7 days post-eclosion, were collected and starved for either 0, 12, or 24 h in 1% agarose (Invitrogen, Göteborg, Sweden) vials and then frozen at −80 °C overnight. To collect hemolymph, 10 flies per replicate were weighed using a 1/10,000 scale (Denver instrument company, Goettingen, Germany), placed in PBS (pH 7.4) in a 1:5 ratio (mg of flies/µL of PBS) and decapitated via centrifugation (at 3000× *g* for 6 min at 4 °C). Hemolymph was used to measure circulating glucose and trehalose. To determine stored trehalose and glycogen, the remaining bodies of the 10 flies were homogenized in a 1:10 ratio of PBS (mg of flies/µL of PBS), homogenates were centrifuged at 12,000× *g* for 15 min at 4 °C, and supernatant was collected for analysis. Trehalose from the hemolymph or supernatant was converted to glucose using porcine kidney trehalase (Sigma, T8778) overnight at 37 °C. Glycogen from the supernatant was converted to glucose by amyloglycosidase from *Aspergillus niger* (Sigma, A7095) overnight at 25 °C. Lastly, glucose levels from all substrates were quantified using the Glucose Assay Kit (Liquick Cor-Glucose Diagnostic kit, Cormay, Siedlce, Poland) involving glucose oxidase and peroxidase according to the manufacturer’s instructions. Briefly, glucose is oxidized to form gluconic acid and hydrogen peroxide by glucose oxidase. The hydrogen peroxide then reacts with 4-aminoantipyrine in the presence of peroxidase to form a colored solution where glucose concentration is proportional to the absorbance of light. Absorbance at 492 nm was measured for each replicate of each substrate on a multi-scan microplate spectrophotometer (model) and converted to a mM concentration of glucose using a linear regression obtained by a calibration curve made from a serial dilution of a sample with a known glucose concentration. Glucose measurements were then converted back to the units of their original substrates by fill-in-the-blank.

### 2.3. Mouse Studies

#### 2.3.1. Subject and Housing Conditions

Adult male C57BL/6 mice (Taconic, Silkeborg, Denmark) weighing approximately 25–30 g at the beginning of the experiment were individually housed in type III standard Macrolon cages in a controlled environment in terms of temperature (20–22 °C) and air humidity (55%) with a 12:12 h light-dark cycle. Water and standard chow (R04, Safe, Augy, France) were available ad libitum at all times unless otherwise stated.

#### 2.3.2. Hmgcr Expression in the Mouse Brain

##### In Situ Hybridization

Design and synthesis of RNA probes: Antisense and sense probes were generated from commercial mouse cDNA clones (Source BioScience). The clones were sequenced (Eurofins MWG Operon, Ebersberg, Germany) and verified to be correct. Plasmids were linearized with appropriate restriction enzymes and the probes were synthesized using a 1 µg vector as a template with T7, Sp6, or T3 RNA polymerase in the presence of digoxigenin (DIG)-labeled 11-UTP (Roche Diagnostics, Rotkreuz, Switzerland). Probes were controlled and quantified using the Nanodrop ND-1000 Spectrophotometer (NanoDrop Technologies, Wilmington, DE, USA).

In situ hybridization on free-floating sections: Free-floating sections were washed in PBS followed by 6% hydrogen peroxide treatment at room temperature. After successive washing in PBS, the sections were treated with 20 µg/mL proteinase K (Invitrogen). The sections were post fixed in 4% formaldehyde (Thermo-Fisher Scientific, Sweden) before pre-incubation in hybridization buffer (50% formamide, 5X SCC pH 4.5, 1% SDS, 50 µg/mL tRNA) (Sigma Aldrich, Stockholm, Sweden), 50 µg/mL heparin (Sigma Aldrich, Stockholm Sweden) in PBS. Hybridization of sections in presence of 100 ng antisense probe/mL was performed overnight at 58 °C. As a control for in situ hybridization, the sense probe was used. To remove the unbound probe, the sections were washed with buffer 1 (50% formamide, 2X SSC pH 4.5, and 0.1% Tween-20 in PBS) followed by buffer 2 (50% formamide, 0.2X SSC pH 4.5, and 0.1% Tween-20 in PBS). The sections were incubated in blocking solution (1% blocking reagent) (Roche Diagnostics, Stockholm, Sweden) followed by overnight incubation combined with 1:5000 diluted anti-digoxigenin alkaline phosphates conjugated antibody (Roche Diagnostics Scandinavia, Stockholm, Sweden). Unbound antibody was washed away with sequential washes in 0.1% TBST. The sections were then developed with Fast Red (Roche diagnostics, Stockholm, Sweden). 

#### 2.3.3. Hmgcr Expression in Mouse Brain under Different Feeding Conditions

Feeding experiments: Two feeding experiments were performed in order to determine Hmgcr expression at baseline, during starvation, and in obese mice (*n* = 8 per group). Experiment 1: Hmgcr mRNA expression at baseline and after 24 h of starvation. Food was removed prior to the onset of the dark phase and the mice were decapitated on the next day. Control mice were sacrificed at the same time but had ad libitum access to food. Experiment 2: Hmgcr mRNA expression in obese mice. An additional cohort of male mice was maintained on either regular chow or a high-fat diet in addition to the regular chow for 8 weeks. Bodyweight measured at the start of the experiments did not differ between the two groups, but the mice on the high-fat diet had significantly increased in body weight compared to the controls after the 8 weeks.

RNA isolation and cDNA synthesis: After 8 weeks, the male mice were sacrificed by cervical dislocation within 2 hours of the early dark phase, and the brains were removed and dissected within 10 min. After dissection, the tissue was immersed in RNA-later solution (Ambion, Elk Grove, CA, USA) and kept at room temperature for 1 h. Individual tissue samples were homogenized by Bullet Blender (Next Advance, New York, NY, USA) in RNA-later solution. mRNA was extracted from tissue using Absolutely RNA Miniprep Kit (Agilent Technologies, Santa Clara, CA, USA) by following the included protocol. RNA concentration was determined using a NanoDrop ND-1000 Spectrophotometer (Thermo Fisher Scientific, Wilmington, DE, USA). cDNA was synthesized using the First Strand cDNA Synthesis Kit (Fisher Scientific, Göteborg, Sweden) with random hexamers (Fisher Scientific, Sweden) as primers according to manufacturer’s instructions.

Quantitative real-time PCR: The cDNA was analyzed with quantitative real-time PCR on MyIQ (Bio-Rad Laboratories, Solna, Sweden). All primers were designed with Beacon Designer v4.0 (Premier Biosoft, San Francisco, CA, USA). β-actin was used as housekeeping gene. Each real-time PCR reaction with a total volume of 20 µL contained cDNA synthesized from 25 ng of total RNA: 0.25 µmol/L of each primer, 20 mol/L Tris/HCl (pH 8.4) (Thermo-Fisher Scientific, Sweden), 50 mol/L KCl (Thermo-Fisher Scientific, Sweden), 4 mol/L MgCl_2_ (Thermo-Fisher Scientific, Sweden), 0.2 mol/L dNTP (Thermo-Fisher Scientific, Sweden), SYBR Green (1:50,000) (Thermo-Fisher Scientific, Sweden). Real-time PCR was performed with 0.02 u/L Taq DNA polymerase (Invitrogen, Stockholm, Sweden), running 30 s at 95 °C, followed by 50 cycles of 10 s at 95 °C, 30 s at 59–61 °C, 30 s at 72 °C, and lastly, 5 min at 72 °C and 10 s at 55 °C. All RT-PCR plates had negative controls included for each primer pair and triplicates for each sample. The quantitative RT-PCR experiments were performed twice to confirm the results. The following primers were used: *Actin* F: 5′-CCTTCTTGGGTATGGAATCCTGTG-3′, R: 5′-CAGCACTGTGTTGGCATAGAGG-3′; *Hmgcr* F: 5′-TTGTGCTTGTATCTAATCT-3′, R: 5′-AATCTATCCAAGAAGTGTAT-3′.

#### 2.3.4. Hmgcr Inactivation in the Mouse Brain

Hypothalamic cannula implantation: Male mice were anesthetized in an induction chamber using isoflurane (Baxter, Deerfield, IL, USA), first, at 4% to induce anesthesia, followed by 1.5% for maintenance, and then restrained onto a stereotaxic frame (Stoelting, Wood Dale, IL, USA). The animals subsequently received a subcutaneous injection of Carprofen (2 mg/kg body weight, Astra Zeneca, Södertälje, Sweden) to prevent post-surgical pain and inflammation. The hair was shaved off the heads of the mice, and the skin was sterilized. An incision was made to expose the skull and local anesthesia (Marcain, 5 mg/kg, Apoteket, Stockholm, Sweden) was added directly onto the surgical cut. The skull was wiped clean with sterile cotton swabs, followed by the application of hydrogen peroxide to enhance the visibility of bregma for correct coordinate determination. Target coordinates used were anterior–posterior −1.48 mm, laterally 0.5 mm, and dorsal–ventral −5.5 mm with respect to bregma. A steel guide cannula (26-gauge, 5.5 mm, World Precision Instruments, Friedberg, Germany) was lowered unilaterally by a micromanipulator (Stoelting, Wood Dale, IL, USA) through the cerebral cortex into the ventromedial hypothalamic nucleus. The guide cannula was secured to the skull using dental cement (GC, Tokyo, Japan). A dummy cannula (26-gauge, 6.0 mm, World Precision Instruments, Friedberg, Germany) was inserted into the guide cannula to prevent clogging and entrance of unfamiliar particles. After surgery, the animals were returned to their home cages and allowed to recover for at least one week prior to further experiments.

Hypothalamic injection of simvastatin: A 5 μL Hamilton syringe (World Precision Instruments, Friedberg, Germany) was attached to an internal cannula (26-gauge, 6 mm, World Precision Instruments, Friedberg, Germany) through polyethylene tubing. The internal cannula was inserted into the guide cannula extending 0.5 mm below the guide cannula tip. Infusions were made 1 h prior to the onset of the dark phase. The mice were mildly anesthetized through isoflurane inhalation and 0.5 μL of simvastatin (Sigma Aldrich, Malmö, Sweden) dissolved in 5% dimethyl sulfoxide (DMSO, Sigma Aldrich) or vehicle (5% DMSO) were infused manually at an approximate speed of 1 µL/min. After the injection, the needle was left in place for 30 s before removal of the internal cannula. Afterward, the mice were returned to their home cages. 

Effects of simvastatin on nocturnal food intake in ad libitum-fed mice: One week before the experiment, the mice were habituated to eating food from Petri dishes. On the day of the experiment, the mice were injected with either simvastatin or vehicle 1 h prior to the onset of the dark phase. A Petri dish with an excess amount of food (8 g of standard chow) was then presented, and the mice’s food intake was measured manually at 1, 2, 3, 6, 12, and 24 h after injection.

Verification of injection sites: After the completion of the experiments, the mice were sacrificed via transcardial perfusion through the left ventricle with PBS followed by 4% formaldehyde (Histolab, Gothenburg, Sweden) and post-fixed at 4 °C overnight. Methyl blue (Sigma Aldrich) was injected into the guide cannula to confirm correct placement (Appendix A). The brains were cut into 200 μm sections using a Leica VT1000S microtome, and the injection site was checked by visual inspection based on Franklin and Paxinos 2007 [25] and Allen Mouse Brain Atlas 10 [26]. 

### 2.4. Rat Studies

#### 2.4.1. Subject, Housing Conditions and Cannula Implantation 

Male Sprague–Dawley rats (AgResearch, Hamilton, New Zealand) weighing 320–331 g at the beginning of the studies were single-housed in Plexiglas cages in a temperature-controlled facility (21 °C) with light-dark 12:12, turning on the light at 8 am. Water and standard laboratory chow (Sharpes Feed, Carterton, New Zealand) were available ad libitum unless stated otherwise. Animals were anesthetized with ketamine (100 mg/kg) and xylazine (20 mg/kg) and surgically equipped with a 26-gauge cannula (Plastics One) aimed at the lateral cerebral ventricle. Stereotaxic coordinates were as follows: 1.5 mm lateral to the midline, 1.0 mm posterior to bregma, and 3 mm below the surface of the skull. Dental acrylic was applied to secure the cannula to two screws inserted in the skull. The injector extended 1 mm beyond the tip of the guide cannula. The placement was verified after a 10-day recovery period and again, following the completion of the feeding experiments, via intracerebroventricular (ICV) injections of 100 ng angiotensin II (rats that drank less than 7 g of water post injection were excluded) [27]. Simvastatin was acquired from Sigma (Christchurch, New Zealand), and it was dissolved in DMSO just before each experimental trial Intracranial injections were performed using Hamilton syringes in a volume of 2 μL. Injection was delivered within 5 s and the injector remained in place in the guide cannula for an additional 15 s to permit diffusion of the drug from the cannula tip.

#### 2.4.2. Hmgcr Inactivation in the Rat Brain 

Experiment 1: Effect of ICV simvastatin on standard chow intake in overnight-deprived rats. Animals deprived of food overnight were injected at 10:00 with vehicle, 10, 30, or 100 nmol simvastatin ICV (*n* = 6–7/group), and chow was returned to hoppers just after the drug administration. Food intake was measured 1, 2, and 4 h post-injection.

Experiment 2: Effect of ICV simvastatin on overnight standard chow intake in ad libitum-fed rats. Ad libitum-fed animals were injected ICV at 19:45 (15 min before the lights off) with vehicle or 30 nmol simvastatin (30 nmol was established to be the lowest orexigenic dose in Experiment 1; *n* = 7/group). Food present in the hoppers was immediately exchanged for pre-weighed pellets and food intake was measured 1, 2, 4, 6, and 12 h post-injection. 

Experiment 3: Effect of ICV simvastatin on episodic intake of palatable solutions. Our protocol was based on previous studies testing the effects of injectants on the short-term intake of palatable solutions in a no-choice scenario [23,24,25]. Animals were accustomed to receiving one of the palatable solutions for 2 h/day on 2 days (10:00–12:00) prior to the injection experiments to avoid neophobia: 10% sucrose (Sigma), 0.1% saccharin (Sigma), or milk (DGC, Hamilton, New Zealand). On the day of the experiment, animals were injected ICV with vehicle or 30 nmol simvastatin (*n* = 6–7/group) just before the presentation of the solution. Intake was measured 2 h post injection. Chow and water were removed during the 2-h palatable tastant exposure. 

### 2.5. Statistical Analysis

Shapiro–Wilk or Kolmogorov–Smirnov normality tests were performed for all data according to the number and type of samples. For data with a normal distribution, one-way ANOVA was performed, while for data with a non-parametric distribution, the Kruskal–Wallis test was used, and in both cases, post hoc tests were used as appropriate for each case (see figure legends for details on the individual experiments). Mean and standard error from all replicates of each experiment were calculated. All analyses were performed with GraphPad Prism 4. Data with a *p* < 0.05 were considered significant.

## 3. Results

### 3.1. Central Hmgcr Regulates Insulin Expression in Drosophila

To begin, we verified that *Hmgcr* was expressed in the insulin-producing cells (IPCs) (Figure 1A). Starving 5–7 day old male flies for 24 h reduced *Hmgcr* transcription to nearly undetectable levels within the IPCs (Figure 1B). Using quantitative RT-PCR, we validated that starvation significantly reduced *Hmgcr* head expression in males (Figure 1C), whereas maintaining the flies on diets containing various concentrations of macronutrients had no significant effect (Figure 1D). 

Next, we wanted to determine if insulin signaling requires Hmgcr activity within the insulin-producing cells (IPCs) of the PI [14]. Since insulin is known to control *Drosophila* body size during development [28], we knocked down *Hmgcr* expression specifically within the IPCs (*Dilp2-GAL4 > UAS-Hmgcr^RNAi^*) throughout larval development to determine if there was an effect on body size. Notably, raising *Dilp2-GAL4 > UAS-Hmgcr^RNAi^* IPC knockdown males (from now on referred to as *Hmgcr* males, Appendix A) at 25 °C was sufficient to produce flies that were significantly lighter than controls (Figure 1E and Appendix A). Raising *Hmgcr* males at 18 °C, which significantly reduces GAL4 activity [29], was sufficient to rescue the phenotype (Appendix A). Consequently, we analyzed insulin-like peptide (*Ilp2*, *Ilp3*, *Ilp5*) and glucagon-like (*Adipokinetic hormone*, *Akh*) gene expression in adult males. *Ilp2*, *Ilp3,* and *Akh* transcript levels were significantly increased in *Hmgcr* males fed normal lab fly food, which constitutes a high-sugar diet (58 g/dL sugar:12 g/dL protein) (Figure 1F). On the other hand, maintaining *Hmgcr* males on a low-sugar diet (10 g/dL sugar:10 g/dL protein) significantly decreased the transcript levels of *Ilp2* and *Ilp3* compared to controls, while *Akh* transcript levels were normal (Figure 1G). Maintaining wild-type males for 24 h on a high-sugar diet containing the Hmgcr inhibitor fluvastatin was sufficient to increase *Akh* transcript levels (Figure 1H), while 5 days of fluvastatin treatment significantly induced the transcript levels of *Ilp2*, *Ilp3,* and *Akh* (Figure 1I). We also demonstrated that *Thor* (*Drosophila* 4E-BP) expression was induced in fluvastatin fed flies (Figure 1J), representative of reduced insulin signaling [30].

Next, we examined the protein levels of ILP2 and ILP3 in the IPCs, comparing controls and *Hmgcr* males fed either a low-sugar or high-sugar diet. Not surprisingly, in controls, abundant ILP2 expression was observed within the IPCs when flies were fed a high-sugar diet (Figure 2A,B). On the other hand, much less ILP2 protein was visible in the IPCs from control flies maintained on a low-sugar diet (Figure 2A,B). In *Hmgcr* males, fed either a high-sugar or low-sugar diet, DILP2 expression was only observed in a few IPCs (Figure 2A,B). A second *Hmgcr* RNAi line showed a similar, although less severe, reduction in DILP2 expression (Appendix A). Apparently, in high-sugar diets, HMGCR knockdown flies increase transcripts of insulin-like peptides (Figure 1F), but for some unknown reason, their translation does not occur (Figure 2A). Interestingly, knocking down *Farnesyl pyrophosphate synthase* (*Fpps*), which is downstream of Hmgcr in the mevalonate pathway, also significantly reduced the expression of DILP2 in the IPCs (Appendix A). In controls, there was no difference in DILP3 expression when comparing flies fed either a high-sugar or low-sugar diet; yet in *Hmgcr* males, no DILP3 protein was observed under either condition (Figure 2C,D).

### 3.2. Central Hmgcr Regulates Insulin Signaling in Drosophila

To clarify if the loss of IPC *Hmgcr* has a direct effect on insulin signaling, we performed Western blot analysis to examine phospho-AKT (pAKT), a key molecule in the insulin signaling pathway, in the heads of flies fed a high-sugar diet [31]. Compared to controls, pAKT levels were significantly reduced in *Hmgcr* males fed a high-sugar diet (Figure 3A,B), indicating a reduction in insulin signaling. It was reported that reduced insulin signaling protects flies against the effects of reactive oxygen species (ROS) [23]. Therefore, we measured the resistance to paraquat, which increases cellular ROS levels, using flies maintained on the two different diets. *Hmgcr* males maintained on a high-sugar diet survived on paraquat-containing food significantly longer than controls (Figure 3C), while those maintained on a low-sugar diet were not significantly different from controls (Figure 3D).

### 3.3. Central Hmgcr Regulates Triglyceride and Carbohydrate Levels in Drosophila

Considering that insulin signaling regulates triglyceride and carbohydrate levels in *Drosophila* [32], we measured their concentrations before and during starvation in adult flies maintained on either a high-sugar or low-sugar diet. *Hmgcr* males maintained on a high-sugar diet or fed a high-sugar diet and then starved for 12 h, had significantly increased triglyceride concentrations compared to controls (Figure 4A). On the other hand, *Hmgcr* males maintained on a low-sugar diet had significantly lower triglyceride concentrations than controls when fed ad libitum (Figure 4A). Similarly, *Hmgcr* males fed a high-sugar diet had elevated circulating glucose concentrations, which remained high after 12 h of starvation (Figure 4B). Again, feeding *Hmgcr* males a low-sugar diet rescued the phenotype (Figure 4B). Moreover, circulating trehalose concentrations were elevated prior to starvation in *Hmgcr* males fed a high-sugar diet, but not in *Hmgcr* males maintained on a low-sugar diet (Figure 4C). Glycogen concentrations were not significantly different from controls on either diet (Figure 4D).

### 3.4. Central Hmgcr Regulates Food Intake in Drosophila

Previously, it was determined that insulin signaling in *Drosophila* regulates feeding behavior [33,34]; therefore, we assessed food intake in *Hmgcr* males. Interestingly, *Hmgcr* males maintained on a high-sugar diet were hyperphagic (Figure 5A), while *Hmgcr* males fed a low-sugar diet showed normal food consumption (Figure 5A). In support of these results, systemic inhibition of Hmgcr, by feeding wild-type flies a high-sugar diet containing fluvastatin, also induced hyperphagia (Figure 5B); while wild-type flies maintained on a low-sugar diet containing fluvastatin ate normally (Figure 5B). Of note, loss of *Hmgcr* expression in the *corpus allatum*, a peripheral endocrine gland where the Hmgcr enzyme is highly active [35,36] and shown to be regulated by insulin [14], had no effect on food intake (Figure 5C). 

*Hmgcr* males fed a high-sugar diet have a reduced insulin response; to substantiate this, we examined the expression of an insulin regulated α-glucosidase gene, known as *target of brain insulin* (*tobi*), whose expression was shown to be inhibited by increasing sugar levels [19]. When *Hmgcr* males were maintained on a high-sugar diet *tobi* expression was reduced compared to similarly feed controls (Figure 5D). On the other hand, feeding control flies a low-sugar diet increased *tobi* expression and in *Hmgcr* males fed a low-sugar diet *tobi* expression was similar to controls (Figure 5D). When maintained on a high-sugar diet, flies where *tobi* was knocked down in the midgut (*48Y-GAL4 > tobi^RNAi^*) were hyperphagic and inhibiting Hmgcr activity with fluvastatin did not increase total food intake (Figure 5E). Yet, overexpressing *tobi* (*48Y-GAL4 > tobi^OE^*) in the midgut significantly inhibited food intake, and feeding these flies fluvastatin increased intake to normal levels (Figure 5F).

To learn more about how Hmgcr regulates food intake, RNA-seq analysis was performed to identify changes in the expression of genes that regulate food intake in Hmgcr males. Our analysis revealed that only three genes, CrzR, CG10477, and Akh, had a significant change in Hmgcr males, all of them decreasing their expression (Table 1). 

### 3.5. Downregulation of CNS Hmgcr Leads to Increased Food Consumption in Rodents

To understand if the novel finding that IPC Hmgcr activity regulates food intake in flies is conserved in mammals, we carried out different experiments in rodents (mice and rats). In mammals, it is known that peripheral insulin signaling to the hypothalamus regulates feeding behavior [32,37]. Thus, we first performed RNA in situ hybridization on adult mouse brains to map Hmgcr expression. Hmgcr was prominently expressed in all major brain areas, including the cortex, amygdala, hippocampus, and hypothalamus (Figure 6A–L, Table 2). Within the hypothalamus, Hmgcr was especially abundant in areas known to regulate food intake, such as the ventromedial nucleus of the hypothalamus (VMH) (Figure 6G) and the arcuate nucleus (ARC) (Figure 6K). 

Next, we sought to downregulate central Hmgcr activity by injecting simvastatin into the mouse hypothalamus (see Appendix A for injection sites), then measuring its acute effect on feeding behavior. A single injection of 150 nmol simvastatin into the mouse hypothalamus had a moderate and transient effect on ad libitum food intake, which appeared 3 h after injection (Figure 6M). Similar to flies, administration of simvastatin peripherally had no effect on feeding behavior in mice (Figure 6N). In addition, consistent with the fly data, Hmgcr expression in the mouse hypothalamus was significantly decreased after 24 h of food deprivation (Figure 6O). 

Next, using rats, we looked further into the effect of downregulating hypothalamic Hmgcr activity on food intake. ICV administration of simvastatin at 30 and 100 nmol significantly increased standard chow intake following overnight food deprivation (Figure 7A). Similar to mice, a moderate and transient increase in food consumption also occurred at 3 h post-injection in ad libitum-fed rats (Figure 7B). On the other hand, ICV simvastatin had no effect on food palatability-driven intake. We gave non-deprived ani-mals episodic access to palatable 0.1% saccharin, 10% sucrose or milk solutions (i.e., a non-carbohydrate sweet tastant, a sweet carbohydrate and a nutritionally complex palatable tastant—all of them being either devoid of energy or energy-dilute) and saw no effect of simvastatin on consumption stimulated by pleasant taste (Figure 7C).

## 4. Discussion

Our findings allow us to present a possible model for how HMGCR links to BMI maintenance, as well as how statins could interfere with this maintenance. Using the model system *Drosophila melanogaster*, we demonstrate that central Hmgcr activity, via the mevalonate pathway, regulates insulin signaling, leading to increased lipid storage, hyperglycemia, and hyperphagia and that this regulation is dependent on carbohydrate consumption. The hyperphagia phenotype was recapitulated in rodents fed a normal diet, where statin inhibition of Hmgcr activity in the hypothalamus led to increased neuronal activity in regions known to regulate food intake. These results provide a strong argument for continued studies of the influence of central HMGCR activity on energy metabolism and food intake as a mechanism for its involvement in the BMI maintenance.

The insulin-glucagon system is highly conserved between flies and mammals [28,30,31,34]. In flies, the insulin system is responsible for regulating energy metabolism and feeding behavior through the insulin-producing cells (IPCs) located in the brain, central for maintaining energy homeostasis [38,39]. We demonstrate that knocking down *Hmgcr* expression in the central IPCs results in increased circulating glucose and increased triglyceride levels, as well as decreased insulin signaling, similar to an insulin resistance-like state in mammals. All the phenotypes resulting from suppression of *Hmgcr* expression in the IPCs (e.g., hyperphagia, hypoglycemia, and increased lipid levels) were induced by a high-sugar diet and returned to the normal state when flies were maintained on low-sugar food (equal parts carbohydrate and yeast). In flies, insulin is known to regulate feeding through its interactions with octopaminergic, dopaminergic, and Neuropeptide F (NPY in humans) signaling [40,41]. In fact, loss of the *Insulin-like receptor* (*InR*) in NPF neurons was sufficient to increase food intake when flies were fed a normal lab high-sugar diet [41]. In that study, the authors did not investigate the effect of feeding the flies different diets. We determined that, on a high-sugar diet, loss of *Hmgcr* expression inhibits insulin signaling, which leads to increased food intake. This could mean that when flies are fed a high-sugar diet, insulin is required to signal to NPF neurons to inhibit overeating. On the other hand, on a less energetic low-sugar diet, insulin signaling may be dispensable. 

We also studied the effect of IPC Hmgcr on the expression of the conserved insulin-regulated α-glucosidase, *target of brain insulin* (*tobi*). *Tobi* expression is regulated by both the insulin-like peptides (ILPs) and the fly glucagon analogue Adipokinetic hormone (AKH). In their screen to identify genes regulated by adult insulins, Buch et al. [19] discovered that loss of the ILPs leads to a significant reduction in *tobi* expression. In fact, *tobi* was the gene most affected by ILP loss [19]. Interestingly, even though the ILPs were required for *tobi* expression, high circulating sugar levels inhibited *tobi*, while high levels of protein increased *tobi* expression. They went on to show that this increase in *tobi* expression required AKH signaling. In our study, we confirm that higher levels of carbohydrate intake inhibit *tobi* expression. Interestingly, similar to ILP loss, inhibiting *Hmgcr* expression in the IPCs reduced *tobi* levels even further (see Figure 5D). In mammals, glycogen metabolism can regulate feeding behavior, where inhibition of glycolysis increases feeding [42,43]. Therefore, through its glycogen regulating activity, Tobi could control feeding in *Drosophila*. In support of this, flies, where *tobi* was knocked down in the midgut, were hyperphagic, and inhibiting Hmgcr activity via fluvastatin did not increase total intake (see Figure 3C). On the other hand, overexpressing *tobi* significantly inhibited food intake, and feeding these flies fluvastatin increased intake back to normal levels (see Figure 3D). Moreover, *Hmgcr* males maintained on a low-sugar diet, where *tobi* levels are normal, were not hyperphagic. Thus, it seems Tobi could control feeding behavior downstream of central Hmgcr activity. 

Many mechanisms controlling feeding behavior and energy balance are conserved between flies and mammals [44,45]. Importantly, in our study, we found that inhibition of Hmgcr activity in the fly hypothalamus-like structure (*pars intercerebralis*) or rodent (mouse and rat) hypothalamus led to a hyperphagic phenotype. In flies, this was most likely due to inhibition of insulin signaling, which has previously been shown to regulate food intake [32,33]. In mice, we demonstrate that Hmgcr is highly expressed in the arcuate nucleus (ARC). Moreover, inhibiting Hmgcr in the rat hypothalamus, via statin injection, led to increased neuronal activity in the ARC and paraventricular nucleus (PVN). Interestingly, the ARC contains two classes of insulin-regulated neurons, the orexigenic neuropeptide Y/agouti-related protein (NPY/AgRP) neurons and anorexigenic pro-opiomelanocortin/cocaine and amphetamine-regulated transcript (POMC/CART) neurons, that signal to the PVN (reviewed by [46,47]).

Overall, manipulating *Hmgcr* gene expression in the brain by both, genetic tools, or statin drugs, might explain in large part the link between HMGCR and the susceptibility to obesity by increased food intake (high energy intake) and fat storage, possibly by causing insulin resistance peripherally and within the brain. Further studies are needed to tease out how central Hmgcr activity, especially within the hypothalamus, regulates feeding behavior.

## 5. Conclusions

This study presents evidence of how the central regulation of Hmgcr can modify metabolism and food intake and, therefore, could explain, to a certain extent, how it influences BMI.

## Figures and Tables

**Figure 1 cells-11-00970-f001:**
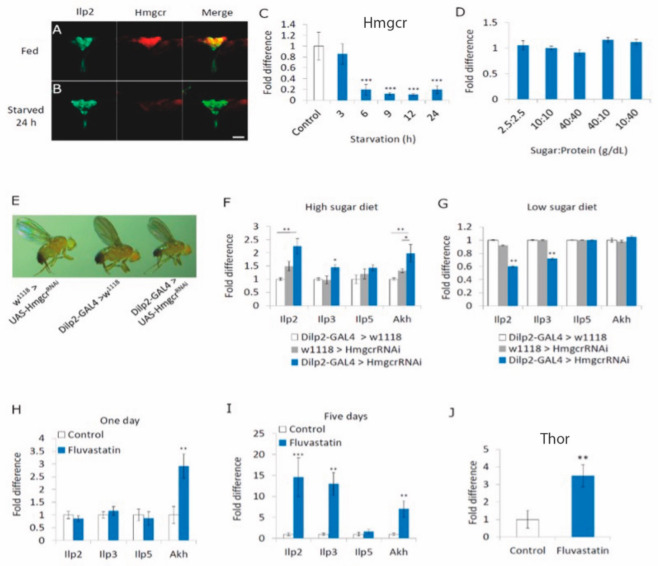
*Hmgcr* expression in the insulin-producing cells is regulated by starvation. (**A**,**B**) Immunohistochemical staining of brains from either equally aged (**A**) normal fed or (**B**) 24 h starved *Drosophila* adult males (5–7 days old). Insulin-like peptide 2 (ILP2) (green) and β-galactosidase in an *Hmgcr* expression pattern (red); yellow indicates overlapping expression (**A**,**B**): *n* = 10 males per parameter, size bar = 50 µm). (**C**) To examine how starvation affects *Hmgcr* transcript levels, RNA was extracted from whole flies fed a high-sugar diet, as well as after various times of starvation. (**D**) To examine how the nutritional state affects *Hmgcr* transcript levels, RNA was extracted from equally aged 5–7 day old male heads under different nutritional states. (**C**,**D**): Flies fed normal food ad libitum were set as 100%, represented by 1 on the graphs. *n* = 10 replicates, each replicate consisted of 25 male fly heads. To detect the significant difference between groups a Shapiro–Wilk test was performed to determine normality, then a one-way ANOVA with Tukey’s post hoc test for multiple comparisons was performed, *** *p* < 0.005). (**E**) Equally aged controls (*w^1118^ > Hmgcr^RNAi^*, *Dilp2-GAL4 > w^1118^*) or where *Hgmcr* is knocked down in the *pars intercerebralis* IPCs (*Dilp2-GAL4 > Hmgcr^RNAi^*), showing that the *Hmgcr* males are smaller than equally aged control flies. This experiment was repeated five times, with each replicate consisting of 10 males (5–7 days old) per genotype. (**F**) Equally aged controls and *Hmgcr* males maintained on a high-sugar diet (58 g/dL sugar:12 g/dL protein) were collected and processed for qPCR. (**G**) Equally aged controls and *Hmgcr* males, 5–7 days old, maintained on a low-sugar diet (10 g/dL sugar:10 g/dL protein) were collected and processed for qPCR. (**H**) Wild-type male flies were fed 0.5 mM fluvastatin for 24 h before being processed for qPCR. (**I**) Wild-type male flies were fed 0.5 mM fluvastatin for 5 days before being processed for qPCR (**F**–**I**): *n* = 10 replicates per genotype, with 25 fly heads per replicate for *Ilp2*, *Ilp3,* and *Ilp5*, or 10 bodies per replicate for *Akh*. To detect the significant difference between groups, a Shapiro–Wilk test was performed to determine normality, then a one-way ANOVA with Tukey’s post hoc test for multiple comparisons was performed, * *p* < 0.05, ** *p* < 0.01). (**J**) Wild-type male flies were fed 0.5 mM fluvastatin for 24 h before being processed for qPCR (*n* = 10 replicates per genotype, with 10 bodies per replicate for Thor). In all graphs, error bars = SEM.

**Figure 2 cells-11-00970-f002:**
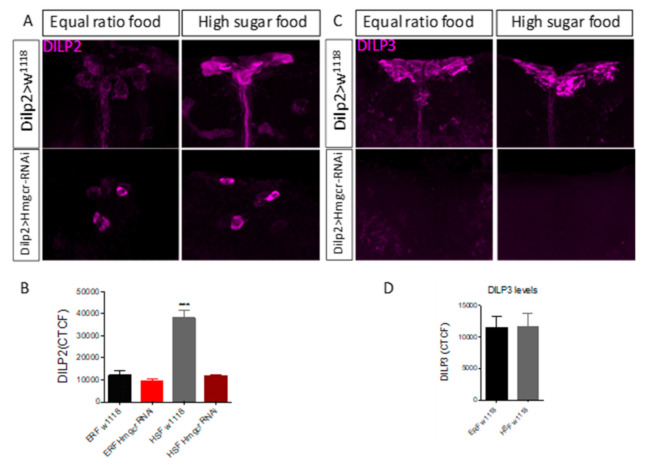
Loss of *Hmgcr* expression in the IPCs affects ILP protein levels. (**A**,**B**) In 5–7 day old control male flies (*Dilp2-GAL4 > w^1118^*), the high-sugar diet (58 g/dL sugar:12 g/dL protein) increased DILP2 protein levels in the *pars intercerebralis* IPCs compared to equally aged male flies fed a low-sugar diet (10 g/dL sugar:10 g/dL protein). In equally aged 5–7 day old *Hmgcr* males (*Dilp2-GAL4 > UAS-Hmgcr^RNAi^*) on either diet, DILP2 protein is expressed in only a few *pars intercerebralis* IPCs. (**C**,**D**) DILP3 protein expression was not affected by either diet in 5–7 day old control male flies, but DILP3 protein was abolished in an equally aged experimental group (*Dilp2-GAL4 > UAS-Hmgcr^RNAi^*) under both conditions. (**B**,**D**) CTCF: corrected total cell fluorescence. In all experiments *n* = 10 male fly heads per genotype per parameter, 5–7 days old. To detect the significant difference between groups a Shapiro–Wilk test was performed to determine normality, then a one-way ANOVA with Tukey’s post hoc test for multiple comparisons was performed, *** *p* < 0.005. In both graphs, error bars = SEM.

**Figure 3 cells-11-00970-f003:**
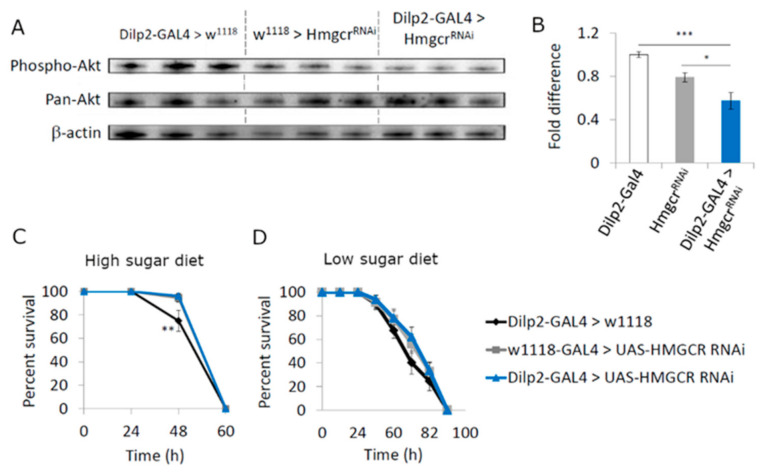
Hmgcr in the insulin-producing cells regulates *Drosophila* insulin signaling. (**A**) Western blot analysis to determine phospho-AKT levels in heads of equally aged adult males (Controls: *Dilp2-GAL4 > w^1118^* and *w^1118^ > UAS-Hmgcr^RNAi^*; Experimental: *Dilp2-GAL4 > UAS-Hmgcr^RNAi^*) maintained on a high-sugar diet (58 g/dL sugar:12 g/dL protein), aged 5–7 days post-eclosion. (**B**) Analysis of phospho-AKT levels compared to pan-AKT (total) present in heads of adult males (Controls: *Dilp2-GAL4 > w^1118^* and *w^1118^ > UAS-Hmgcr^RNAi^*; Experimental: *Dilp2-GAL4 > UAS-Hmgcr^RNAi^*) raised on a high-sugar diet 58 g/dL sugar:12 g/dL protein, aged 5–7 days post-eclosion (A and B: *n* = 10 males per genotype, 5–7 day old. The experiment was repeated five times. In (**B**), statistical analysis between groups involved a Shapiro–Wilk test to determine normality, then a one-way ANOVA with Tukey’s post hoc test for multiple comparisons was performed, * *p* < 0.05, ** *p* < 0.05, *** *p* < 0.005. Error bars = SEM. (**C**,**D**) Equally aged adult males (Controls: *Dilp2-GAL4 > w^1118^* and *w^1118^ > UAS-Hmgcr^RNAi^*; Experimental: *Dilp2-GAL4 > UAS-Hmgcr^RNAi^*) raised on either a (**C**) high-sugar (58 g/dL sugar:12 g/dL protein) or (**D**) low-sugar diet (10 g/dL sugar:10 g/dL protein), aged 5–7 days post-eclosion, were maintained on food containing 20 mM paraquat to determine their survival rate (*n* = 10 flies per genotype per replicate, with five replicates per experiment, the experiment was repeated five times).

**Figure 4 cells-11-00970-f004:**
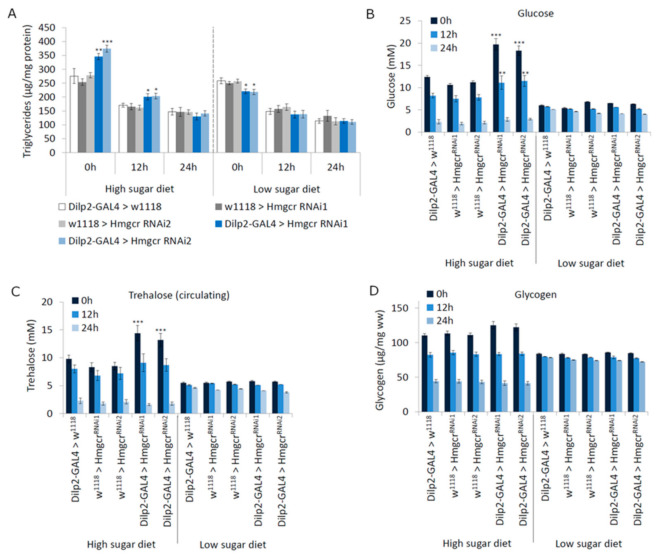
Hmgcr expressed in the insulin-producing cells regulates metabolism. (**A**) Triacylglycerol (TAG) levels were determined in male flies (Controls: *Dilp2-GAL4 > w^1118^* and *w^1118^ > UAS-Hmgcr^RNAi^*; Experimental: *Dilp2-GAL4 > UAS-Hmgcr^RNAi^*) raised on either a high-sugar or low-sugar diet after 0, 12, and 24 h of starvation. (**B**–**D**) Circulating levels of carbohydrates were measured in male flies (Controls: *Dilp2-GAL4 > w^1118^* and *w^1118^ > UAS-Hmgcr^RNAi^*; Experimental: *Dilp2-GAL4 > UAS-Hmgcr^RNAi^*) raised on either a high-sugar or low-sugar diet, aged 5–7 days post-eclosion, (**B**) circulating glucose, (**C**) circulating trehalose, (**D**) stored glycogen. In all graphs, where present: * *p* < 0.05, ** *p* < 0.01, *** *p* < 0.005. To detect the significant difference between groups, a Shapiro–Wilk test was performed to determine normality, then a one-way ANOVA with Tukey’s post hoc test for multiple comparisons; *n* = 300 flies for all strains at each time point for all experiments. In all figures, error bars = SEM.

**Figure 5 cells-11-00970-f005:**
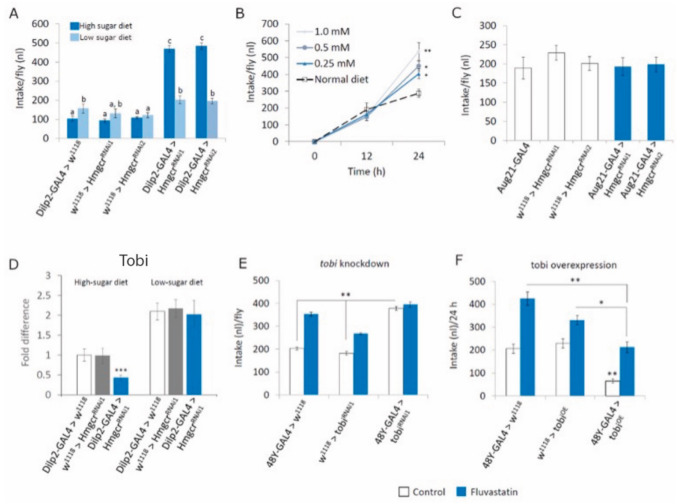
IPC Hmgcr regulates feeding in adult males (**A**) An intake assay [19] was used to assess total food intake in equally aged 5–7 day old adult male flies, fed either a high-sugar or low-sugar diet over a 24 h period. Different letters indicate similar groups (i.e., ‘a’ is significantly different than ‘b’ or ‘c’ and so on. One-way ANOVA with Tukey’s post hoc test for multiple comparisons, *p* < 0.05). (**B**) Adult male flies, 5–7 days old, maintained on either a high-sugar diet, were fed fluvastatin-containing food for 24 h and total food intake was measured every 12 h (**A**,**B**: *n* = 10 replicates with 5 males per replicate. To detect the significant difference between groups, a Shapiro–Wilk test was performed to determine normality, then a one-way ANOVA with Tukey’s post hoc test for multiple comparisons was performed, * *p* < 0.05, ** *p* < 0.01). (**C**) A CAFE assay was used to assess total food intake in flies fed a high-sugar diet over a 24 h period in adult males where *Hmgcr* is specifically knocked down in the *corpus allatum* (*Aug21-GAL4 > Hmgcr^RNAi^*) (As per standard protocols, *n* = 10 replicates with 5 males per replicate, per genotype. To detect the significant difference between groups a Shapiro–Wilk test was performed to determine normality, then a one-way ANOVA with Tukey’s post hoc test for multiple comparisons was performed). (**D**) Controls and *Hmgcr* males were maintained on either a high-sugar or low-sugar diet before being collected and processed for qPCR to determine *tobi* expression levels (*n* = 10 replicates per genotype, 10 whole bodies per sample for *tobi*. To detect the significant difference between groups, a Shapiro–Wilk test was performed to determine normality, then a one-way ANOVA with Tukey’s post hoc test for multiple comparisons was performed. *** *p* < 0.001). (**E**,**F**) *Tobi* was either € knocked down or (**F**) overexpressed in gut endoderm, equally aged 5–7 day old male flies were raised on a high-sugar diet with or without the Hmgcr antagonist fluvastatintin. (**E**) Knocking down *tobi* increased total food intake, this was not affected by fluvastatin, while (**F**) overexpressing *tobi* inhibited total food intake, this was rescued by feeding the files fluvastatin. (In (**E**,**F**), where present, different letters indicate a similar group. *n* = 10 replicates with 5 males per replicate. To detect the significant difference between groups a Shapiro–Wilk test was performed to determine normality, then a one-way ANOVA with Tukey’s post hoc test for multiple comparisons was performed, * *p* < 0.05, ** *p* < 0.01). In all figures, error bars = SEM.

**Figure 6 cells-11-00970-f006:**
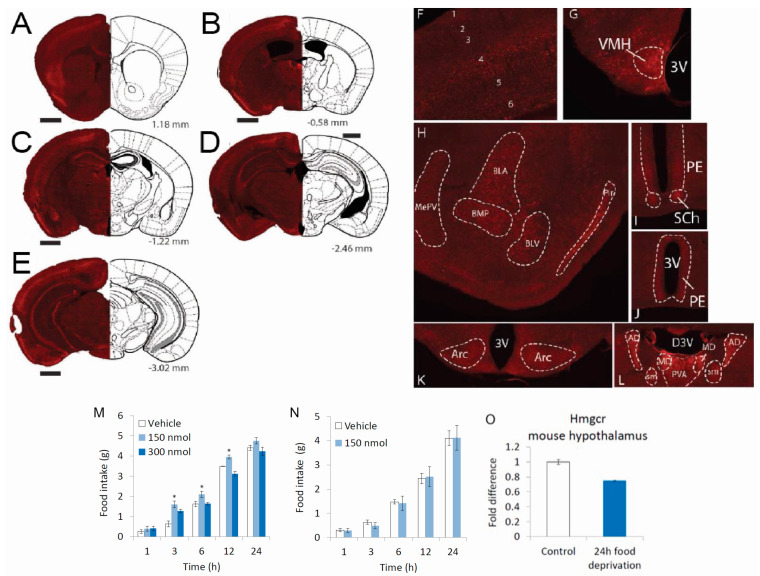
Simvastatin regulates feeding in mice. Expression profile of *Hmgcr* mRNA using fluorescent floating in situ hybridization on coronal brain sections of adult male C57BL/6 mice visualized as overview and detailed pictures. (**A**–**E**): Representative pictures of magnified images of the cortex, hippocampus, thalamus, amygdala, and hypothalamus. (**F**) Cortex layers 1–6 (1–6). (**G**) Basolateral amygdaloid nucleus, anterior part (BLA), basolateral amygdaloid nucleus, ventral part (BLV), basomedial amygdaloid nucleus, posterior part (BMP), dorsal endopiriform claustrum (DEn), ventral endopiriform claustrum (VEn), medial amygdaloid nucleus, posteroventral part (MePV), (**H**) ventromedial hypothalamic nucleus (VMH), (**I**) periventricular hypothalamic nucleus (Pe), suprachiasmatic nucleus (SCh), intercalated nuclei of the amygdala (**I**), (**J**) periventricular hypothalamic nucleus (Pe), third ventricle (3V), (**K**) arcuate hypothalamic nucleus, dorsal part (ArcD), arcuate hypothalamic nucleus, lateral part (ArcL) (**L**) anterodorsal thalamic nucleus (AD), nucleus of the stria medullaris (sm), mediodorsal thalamic nucleus (MD), paraventricular thalamic nucleus, anterior part (PVA), dorsal third ventricle (d3v). Bregma levels and described brain regions are according to Franklin and Paxinos 2007 [25] and Allen Mouse Brain Atlas [26]. Black scale bar, 1 mm. (**M**) Various concentrations of simvastatin were injected directly into the hypothalamus of mice, after which, the total amount of food consumed was measured (*n* = 10 mice per concentration, one-way ANOVA with Bonferroni post hoc test for multiple comparisons, * *p* < 0.05). (**N**) Various concentrations of simvastatin were injected peripherally into the peritoneal of mice, after which the total amount of food consumed was measured (*n* = 10 mice per concentration, one-way ANOVA with Bonferroni post hoc test for multiple comparisons, * *p* < 0.05). (**O**) Relative level of Hmgcr transcript in the hypothalamus from starved male mice (*n* = 10 qPCR runs; one-way ANOVA with Bonferroni post hoc test for multiple comparisons, * *p* < 0.05). In all graphs, error bars = SEM.

**Figure 7 cells-11-00970-f007:**
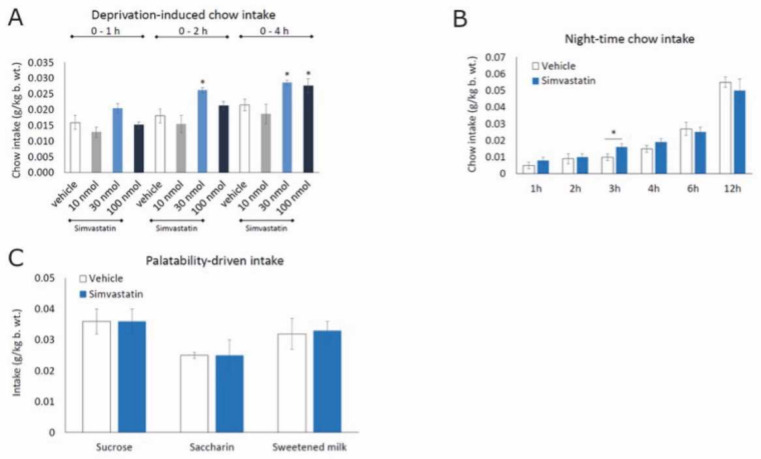
Simvastatin regulates neuronal activity in the hypothalamus. Effect of ICV simvastatin on (**A**) standard chow intake induced by overnight deprivation, (**B**) night-time chow intake, and (**C**) episodic (2 h) intake of palatable, energy-dilute solutions (*n* = 6–7 rats per concentration per time point, one-way ANOVA with Bonferroni post hoc test for multiple comparisons, * *p* < 0.05).

**Table 1 cells-11-00970-t001:** Genes related to food intake changed by Hmgcr inhibition.

FlyBase ID	Gene	Expression	*p*-Value
**FBgn0036278**	**CrzR**	**Downregulated**	**5 × 10^−5^**
**FBgn0035661**	**CG10477**	**Downregulated**	**5 × 10^−5^**
**FBgn0004552**	**Akh**	**Downregulated**	**0.00015**
FBgn0027109	NPF	Downregulated	0.0622
FBgn0261565	Lmpt	Downregulated	0.1465
FBgn0038552	Alg1	Downregulated	0.19025
FBgn0034579	mRpL54	Downregulated	0.20715
FBgn0004889	tws	Downregulated	0.3976
FBgn0039737	CG7929	Downregulated	0.40555
FBgn0053087	LRP1	Downregulated	0.528
FBgn0010905	Spn	Downregulated	0.72525
FBgn000469	Ptp99A	Downregulated	0.9842
FBgn0260660	mp	Upregulated	0.05085
FBgn0085385	Bma	Upregulated	0.2656
FBgn0003861	trp	Upregulated	0.33845
FBgn0261278	Grp	Upregulated	0.54465

**Table 2 cells-11-00970-t002:** Graded levels of *Hmgcr* mRNA expression in the mouse brain. A sign − means that no expression of Hmgcr mRNA was detected by RNA in situ hybridization. The signs +, ++ and +++ mean significant Hmgcr mRNA detection with respect to the areas without detection with *p* < 0.05, *p* < 0.01, *p* < 0.001, respectively.

Site	Expression
**Telencephalon**	
Medal septal nuclei (MS)	−
Ventral pallidum	−
**Amygdala**	
Basolateral amygdaloid nu, anterior (BLA)	+++
Basolateral amygdaloid nu, ventral (BLV)	+++
Basolateral amygdaloid nu, posterior (BLP)	+++
Basomedial amygdaloid nu, anterior (BMA)	+++
Basomedial amygdaloid nu, posterior (BMP)	+++
Central amygdaloid nu, caspular dic (CeC)	++
Central amygdaloid nu, lateral div (CeL)	++
Intercalated nu amygdala (I)	+++
Lateral amygdaloid nu, dorsolateral (LaDL)	++
Lateral amygdaloid nu, ventromedial (LaVM)	+
**Cerebral cortex**	
Dorsal endopiriform nu (DEn)	++
Layer 1 (1)	−
Layer 2 (2)	−
Layer 3 (3)	−
Layer 4 (4)	+++
Layer 5 (5)	+
Layer 6 (6)	++
Piriform cortex (Pir)	+++
Ventral endopiriform nu (VEn)	++
**Hippocampal formation**	
Granule layer, dentate gyrus (GrDG)	++
Lacunosum molecular layer, hipp (LMol)	+
Molecular layer dentate gyrus (Mol)	−
Oriens layers, hippocampus (Or)	++
Polymorph layer, dentate gyrus (PoDG)	+
Pyramidal cell layers, hippocampus (Py)	+
Stratum radiatum, hippocampus (Rad)	−
**Diencephalon**	
**Hypothalamus**	
Arcuate hypothalamic nu, dorsal (ArcD)	+++
Arcuate hypothalamic nu, lateral (ArcL)	+++
Dorsomedial hypothalamic nu (DM)	+
Lateral hypothalamic area (LH)	++
Paraventricular hypothalamic lateral, magnocellular part (PaLM)	++
Paraventricular hypothalamic medial, magnocellular part (PAMM)	+
Periventricular hypothalamic nu (Pe)	++
Posteror hypthoalamic nu (PH)	+
Ventromedial hypothalamic nu, central (VMHC)	++
**Thalamus**	
Angular thalamic nu (Ang)	−
Anterodorsal thalamic nu (AD)	+++
Anteromedial thalamic nu (AM)	+
Anteroventral thalamic nu (AV)	++
Central medial thalamic nu (CM)	+
Dorsal lateral geniculate nu (DLG)	−
Ethmoid thalamic nu (Eth)	−
Intermediodorsal thalamic nu (IMD)	+
Lat post thalamic nu, laterorostral (LPLR)	−
Lat post thalamic nu, medorostral (LPMR)	−
Lateral habenular nu (LHb)	+
Laterodorsal thalamic nu (LD)	+
Mamillotalamic tract (mt)	−
Medial habenular nu (MHb)	+
Medial geniculate nu, dorsal (MGD)	+
Medial geniculate nu, ventral (MGV)	+
Mediodorsal thalamic nu (MD)	++
Nigrostriatal nucleus (ns)	−
Oval paracentral nu (OPC)	−
Paracentral thalamic nu (PC)	+
Parafascicular thalamic nu (PF)	−
Parataenial thalamic nu (PT)	−
Paraventricular thalamic nu (PV)	+
Posterior thalamic nu group (Po)	+
Posteromedian thalamic nu (PoMn)	+
Peticular thalamic nu (Rt)	−
Reuniens thalamic nu (Re)	+
Rhomboid thalamic nu (Rh)	+
Stria medullaris, thalamus (STIA)	+
Submedius thalamic nu (Sub)	+
V posteromed thalamic, nu, parvicel (VPPC)	−
Ventral posterolat thalamic nu (VPL)	+
Ventral posteromed thalamic nu (VPM)	+
Ventral reunions thalamic nu (VRe)	−
Ventrolateral geniculate nu (VLG)	+
Ventrolateral thalamic nu (VL)	+
Ventromed thalamic nu (VM)	−
Zona inserta, dorsal (ZID)	−
Zona inserta, ventral (ZIV)	+
**Mesencephalon**	
Central nu inferior colliculus (CIC)	+
Dorsal cortex, inferior colliculus (DCIC)	−
Dorsal raphe nu, inferior (DRI)	−
Dorsal raphe, caudal part (DRC)	−
Dorsal raphe nu, ventral (DRV)	+
External cortex, inferior colliculus (ECIC)	+
Pons	
Kölliker-Fuse nu (KF)	+
Laterodorsal tegment nu,	−
ventral (LDTgV)	−
Lateral parabrachial nu (LPB)	−
Motor trigmenial nu (Mo5)	+
Ventral tegmental nu (VTg)	+
**Medulla**	
Ambiguus nu (Amb)	+
Area postrema (AP)	+
Dorsal motor nu vagus *n* (10)	−
Fascial nu (7)	−
Gigantocellular reticular nu, alpha (GiA)	+
Hypoglossal nu (12)	−
Inferior olive, beta subnu (IOBe)	−
Inferior olive, cap of Kooy med nu (IOK)	+
Inferior olive, dorsal accessory nu (IOD)	+
Inferior olive, dorsomed cell grp (IODM)	+
Inferior olive, dorsomed cell col (IODMC)	+
Inferior olive, med nu (IOM)	+
Inferior olive, principal nu (IOPr)	−
Inferior olive, subnu B of med nu (IOB)	+
Inferior olive, subnu C of med nu (IOC)	+
Medial vestibular nu (MVe)	−
nu of solitary tract, commissural (SolC)	+
nu of solitary tract, dorsolateral	+
tract (SolDL)	+
nu of solitary tract, medial (SolM)	+
nu of solitary tract, ventrolateral (SolVL)	+
Prepositus hypoglossal nu (Pr)	−
Pyramidal tract (py)	+
Raphe magnus nu (RMg)	+
Raphe obscurus nu (Rob)	+
Raphe pallidus (RPa)	+
Solitary tract (sol)	−
**Cerebellum**	
Granular layer	+
Molecular layer	+
Purkinje cell layer	+

## Data Availability

The authors confirm that the data supporting the findings of this study are available within the article and its Appendix A.

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
