# Peer review of "The Statin Target Hmgcr Regulates Energy Metabolism and Food Intake through Central Mechanisms"

_cells, 2022, doi:10.3390/cells11060970_

Round 1

Reviewer 1 Report

Accept

Author Response

Dear Reviewer,

 We appreciate all your comments that help us improve our work

Reviewer 2 Report

The authors investigated the role of 3-hydroxy-3-methylglutaryl-CoA reductase (HMG-CoA, also Hmgcr) expressed by the brain in energy metabolism using flies, mice, and rats. In flies, they found that Hmgcr is strongly expressed by the brain. Knockdown of the Hmgcr in the insulin-producing cells (IPCs) of the brain resulted in an increase in insulin-like peptides under high sugar diet feeding, which was reproduced by the treatment with the Hmgcr inhibitor Fluvastatin, but decreased under a low sugar diet condition. pAKT phosphorylation was also reduced in the heads when Hmgcr in IPCs was knocked down. Knockdown of Hmgcr in IPCs decreased body size, induced lipid storage, hyperglycemia, and hyperphagia, the last of which was mediated via the target of brain insulin (tobi). In mice, they determined the neuroanatomical distribution of Hmgcr mRNA in the brain. Hypothalamic injection of simvastatin, an inhibitor of Hmgcr, increased food intake. Intracerebroventricular (ICV) injections of simvastatin stimulated food intake in rats. Based on these data, the authors concluded that central Hmgcr plays a role in controlling energy balance and food intake.

This manuscript demonstrates some interesting data. In particular, the novelty of this work lies in providing evidence demonstrating that brain HMG-CoA controls food intake and energy balance. While some of the data are sound and convincing and constitute an interesting body of work, most studies are descriptive rather than explanatory. A major concern I have with this manuscript is that the conclusion may not be fully backed by the current set of data.

Although the authors concluded that “inhibiting mevalonate pathway genes, including Hmgcr and Farnesyl pyrophosphate synthase (Fpps), in insulin-producing cells of the Drosophila pars intercerebralis (PI), the fly hypothalamic equivalent, significantly reduces the expression of insulin-like peptides” (page 1, line 21-23), the data presented in this manuscript (Fig 1F and G) also show that, under high sugar conditions, insulin-like peptides are rather increased by either knockdown of Hmgcr or its inhibitor. Need more explanation.

Along the same line, the following conclusion should be supported by more rigorous data. “In rats and mice, acute inhibition of hypothalamic Hmgcr activity stimulates food intake, as well as enhances neuronal activity within the rat arcuate and paraventricular nuclei.” (page 1, line 26-27). My concern is why higher doses (300 nmol in Fig 6M) of simvastatin did not induce any effect on food intake while having a significant effect with the lower dose of 150 nmol.  If this effect is truly via the inhibition of HMG-CoA, the higher doses are used, the more inhibition of this enzyme should occur and the clearer effect is expected in general. I think it would be appropriate to measure brain HMG-CoA activity to determine if this manipulation really inhibits the enzyme.  This is also the case in Fig7A. In addition, I could not find the data directly supporting the conclusion of “enhances neuronal activity within the rat arcuate and paraventricular nuclei.” Probably, c-fos staining is the most common and appropriate method to show neural activation in the brain in vivo.

I find myself difficult to follow the logical flows in this manuscript. For example, reference (10) was cited for “Hmgcr expression requiring insulin signaling”. However, this paper is an epidemiological study reporting that statins increase the risk of type 2 diabetes and I did not find the claim in the paper. Please cite the appropriate paper. In a similar vein, reference 20 does not support the authors’ claim. The authors assumed the reduced insulin signaling may have the effect on paraquat-induced oxidative toxicity, based on Ref 20 reporting that reduced insulin signaling protects flies against the effects of reactive oxygen species (ROS). However, Ref. 20 did not provide any links between insulin and ROS.

Minors.

  1. What is the control for the in situ hybridization of Hmgcr mRNA?
  2. is there a link between increased insulin-like peptides production, reduced pAKT, increased lipid storage, and hyperglycemia ?
  3. Please quantify the size of the flies in Fig 1E.
  4. Please present larger images of in situ data (Fig. 6A-L). The current version is too small.

Author Response

Dear Reviewers,

We appreciate all your comments that help us improve our work, below you will find answers to all your comments in blue and bold letters.

The authors investigated the role of 3-hydroxy-3-methylglutaryl-CoA reductase (HMG-CoA, also Hmgcr) expressed by the brain in energy metabolism using flies, mice, and rats. In flies, they found that Hmgcr is strongly expressed by the brain. Knockdown of the Hmgcr in the insulin-producing cells (IPCs) of the brain resulted in an increase in insulin-like peptides under high sugar diet feeding, which was reproduced by the treatment with the Hmgcr inhibitor Fluvastatin, but decreased under a low sugar diet condition. pAKT phosphorylation was also reduced in the heads when Hmgcr in IPCs was knocked down. Knockdown of Hmgcr in IPCs decreased body size, induced lipid storage, hyperglycemia, and hyperphagia, the last of which was mediated via the target of brain insulin (tobi). In mice, they determined the neuroanatomical distribution of Hmgcr mRNA in the brain. Hypothalamic injection of simvastatin, an inhibitor of Hmgcr, increased food intake. Intracerebroventricular (ICV) injections of simvastatin stimulated food intake in rats. Based on these data, the authors concluded that central Hmgcr plays a role in controlling energy balance and food intake.

This manuscript demonstrates some interesting data. In particular, the novelty of this work lies in providing evidence demonstrating that brain HMG-CoA controls food intake and energy balance. While some of the data are sound and convincing and constitute an interesting body of work, most studies are descriptive rather than explanatory. A major concern I have with this manuscript is that the conclusion may not be fully backed by the current set of data.

Q1.

Although the authors concluded that “inhibiting mevalonate pathway genes, including Hmgcr and Farnesyl pyrophosphate synthase (Fpps), in insulin-producing cells of the Drosophila pars intercerebralis (PI), the fly hypothalamic equivalent, significantly reduces the expression of insulin-like peptides” (page 1, line 21-23), the data presented in this manuscript (Fig 1F and G) also show that, under high sugar conditions, insulin-like peptides are rather increased by either knockdown of Hmgcr or its inhibitor. Need more explanation.

R1. You are right, the levels of insulin-like peptide transcripts increase with a high-sugar diet, but as you can see in figure 2, the protein decreases, which indicates that for some reason its translation does not take place. This has now been clarified better in the manuscript.

In Hmgcr males, fed either a high-sugar or low-sugar diet, DILP2 expression was only observed in a few IPCs (Figure 2 A,B). A second Hmgcr RNAi line showed a similar, although less severe, reduction in DILP2 expression (Figure S3 A,C).Apparently, in high-sugar diets, HMGCR knockdown flies increase transcripts of insulin-like peptides (Figure 1F), but for some unknown reason, their translation does not occur (Figure 2A).

Q2.

Along the same line, the following conclusion should be supported by more rigorous data. “In rats and mice, acute inhibition of hypothalamic Hmgcr activity stimulates food intake, as well as enhances neuronal activity within the rat arcuate and paraventricular nuclei.” (page 1, line 26-27). My concern is why higher doses (300 nmol in Fig 6M) of simvastatin did not induce any effect on food intake while having a significant effect with the lower dose of 150 nmol.  If this effect is truly via the inhibition of HMG-CoA, the higher doses are used, the more inhibition of this enzyme should occur and the clearer effect is expected in general. I think it would be appropriate to measure brain HMG-CoA activity to determine if this manipulation really inhibits the enzyme.  This is also the case in Fig7A. In addition, I could not find the data directly supporting the conclusion of “enhances neuronal activity within the rat arcuate and paraventricular nuclei.” Probably, c-fos staining is the most common and appropriate method to show neural activation in the brain in vivo.

R2. We agree with the referee, we do not know the reason why higher doses of simvastatin do not produce a greater effect in mammals. This could perhaps be due to off-target effects, and it could be interesting to analyze in future studies. Regarding the conclusion "enhances neuronal activity within the rat arcuate and paraventricular nuclei", as you mention, the present work does not have enough evidence and therefore we decided to omit it from the manuscript.

Q3.

I find myself difficult to follow the logical flows in this manuscript. For example, reference (10) was cited for “Hmgcr expression requiring insulin signaling”. However, this paper is an epidemiological study reporting that statins increase the risk of type 2 diabetes and I did not find the claim in the paper. Please cite the appropriate paper. In a similar vein, reference 20 does not support the authors’ claim. The authors assumed the reduced insulin signaling may have the effect on paraquat-induced oxidative toxicity, based on Ref 20 reporting that reduced insulin signaling protects flies against the effects of reactive oxygen species (ROS). However, Ref. 20 did not provide any links between insulin and ROS.

R3. You are right, so we checked that all the references were adequate.

Minors

Q4.

  1. What is the control for the in situ hybridization of Hmgcr mRNA?

R4. Your question is very appropriate, we used the sense probe as a control for in situ hybridization and this was added in the manuscript as follows:

Free-floating sections were washed in PBS followed by 6% hydrogen peroxide treatment at room temperature. After successive washing in PBS, the sections were treated with 20 µg/ml proteinase K (Invitrogen). The sections were post-fixed in 4% formaldehyde (Thermo-Fisher Scientific, Sweden) before pre-incubation in hybridization buffer (50% formamide, 5x SCC pH 4.5, 1% SDS, 50 µg/ml tRNA) (Sigma Aldrich, Stockholm, Sweden), 50 µg/ml heparin (Sigma Aldrich, Stockholm Sweden) in PBS. Hybridization of sections in presence of 100 ng antisense probe/ml was performed overnight at 58 °C. As a control for the in situ hybridization, the sense probe was used. To remove unbound probe, the sections were washed with buffer 1 (50% formamide, 2x SSC pH 4.5 and 0.1% Tween-20 in PBS) followed by buffer 2 (50% formamide, 0.2x SSC pH 4.5 and 0.1% Tween-20 in PBS). The sections were incubated in blocking solution (1% blocking reagent) (Roche Diag-nostics, Sweden, Stockholm) followed by overnight incubation combined with 1:5,000 diluted anti-digoxigenin alkaline phosphates conjugated antibody (Roche Diagnostics Scandinavia, Stockholm, Sweden). Unbound antibody was washed away with sequential washes in 0.1% TBST. The sections were then developed with Fast Red (Roche diagnostics, Stockholm, Sweden).

Q5.

  1. is there a link between increased insulin-like peptides production, reduced pAKT, increased lipid storage, and hyperglycemia ?

R5. What you mention occurs in insulin resistance. This was added to the manuscript.

We demonstrate that knocking down Hmgcr expression in the central IPCs results in increased circulating glucose and increased triglyceride levels, as well as decreased insulin signaling, similar to an insulin resistance-like state in mammals.

Q6.

  1. Please quantify the size of the flies in Fig 1E.

R6. The size of the flies did not show easily quantifiable differences, but their weight did. This has been corrected in manuscript and weight data appear in Supplementary Fig. 2B.

Supplemental Figure 2. Loss of Hmgcr expression in Dilp2 cells reduces adult fly weight. (A) Two different Hmgcr UAS-RNAi lines were crossed to the nervous system driver elav-GAL4 and equally aged 5-7 day old adult males were used for qPCR analysis to determine if the RNAi lines were functional. (UAS-HmgcrRNAi1 = P{KK101807}sVIE-260B, UAS-HmgcrRNAi2 = P{UAS-RNAi-HMGCR}our10367-R3). (n = 10 replicates, consisting of 10 male whole bodies, 5-7 days old). Different letters indicate similar groups (i.e. ‘a’ is significantly different than ‘b’ or ‘c’ and so on, one-way ANOVA with Tukey’s post hoc test for multiple comparisons was performed). Error bars = SEM. (B) Hmgcr was knocked down in the insulin-producing cells, the average body weight of adult male flies decreased in the experimental group (Dilp2-Gal4>UAS-HmgcrRNAi) compared to the controls (Dilp2-Gal4>w1118 and w1118>UAS-HmgcrRNAi) when reared at 25 oC. When reared at 18 oC, there was no difference in the body weight in both experimental and control groups. (n=10 for each group, One-Way ANOVA with a Bonferroni posthoc test was performed to detect the significance. *** P < 0.001) Error bars = SEM.

Q7.

  1. Please present larger images of in situ data (Fig. 6A-L). The current version is too small.

R7. As you mentioned, we increased the size of the in situ hybridization images. They now appear in the manuscript as follows:

Figure 6. Simvastatin regulates feeding in mice. Expression profile of Hmgcr mRNA using fluorescent floating in situ hybridization on coronal brain sections of adult male C57BL/6 mice visualized as overview and detailed pictures. (A-E: Representative pictures of magnified images of the cortex, hippocampus, thalamus, amygdala, and hypothalamus (F) Cortex layers 1-6 (1-6). (G) Basolateral amygdaloid nucleus, anterior part (BLA), basolateral amygdaloid nucleus, ventral part (BLV), basomedial amygdaloid nucleus, posterior part (BMP), dorsal endopiriform claustrum (DEn), ventral endopiriform claustrum (VEn), medial amygdaloid nucleus, posteroventral part (MePV), (H) ventromedial hypothalamic nucleus (VMH), (I) periventricular hypothalamic nucleus (Pe), suprachiasmatic nucleus (SCh), intercalated nuclei of the amygdala (I), (J) periventricular hypothalamic nucleus (Pe), third ventricle (3V), (K) arcuate hypothalamic nucleus, dorsal part (ArcD), arcuate hypothalamic nucleus, lateral part (ArcL) (L) anterodorsal thalamic nucleus (AD), nucleus of the stria medullaris (sm), mediodorsal thalamic nucleus (MD), paraventricular thalamic nucleus, anterior part (PVA), dorsal third ventricle (d3v). Bregma levels and described brain regions are according to Franklin and Paxinos 2007 [60] and Allen Mouse Brain Atlas [61]. Black scale bar, 1mm. (M) Various concentrations of simvastatin were injected directly into the hypothalamus of mice, after which the total amount of food consumed was measured (n = 10 mice per concentration, one-way ANOVA with Bonferroni post hoc test for multiple comparisons, * P < 0.05). (N) Various concentrations of simvastatin were injected peripherally into the peritoneal of mice, after which the total amount of food consumed was measured (n= 10 mice per concentration, one-way ANOVA with Bonferroni post hoc test for multiple comparisons, * P < 0.05). (O) Relative level of Hmgcr transcript in the hypothalamus from starved male mice (n = 10 qPCR runs; one-way ANOVA with Bonferroni post hoc test for multiple comparisons, * P < 0.05). In all graphs error bars = SEM

Round 2

Reviewer 2 Report

The authors have satisfactorily addressed my concerns.